# Effects and Underlying Mechanisms of Zearalenone Mycotoxin at Concentrations Close to the EC Recommendation on the Colon of Piglets after Weaning

Valeria Cristina Bulgaru [1,2,*], Ana Maria Pertea [1], Iulian Alexandru Grosu [1], Andrei Cristian Anghel [1], Gina Cecilia Pistol [1], Daniela Eliza Marin [1], Anca Dinischiotu [2] and Ionelia Taranu [1]

1   Laboratory of Animal Biology, National Institute of Research and Development for Biology and Animal Nutrition, 077015 Balotesti, Romania
2   Faculty of Biology, University of Bucharest, Splaiul Independentei 91-95, 050095 Bucharest, Romania
*   Correspondence: cristina.bulgaru@ibna.ro

**Abstract:** Zearalenone (ZEN) is a mycotoxin produced by *Fusarium* fungi that contaminates food and feed, affecting both human and animal health. Among farm animals, the pig is a great consumer of grains and has a native sensitivity to mycotoxins. As the main route of contamination is oral, the intestine is the first defense barrier that plays an important role in the immune response being able to secrete effector molecules (cytokines). At the European level, there are no regulations regarding the amount of ZEN that can be present in the feed of piglets, only recommendations for piglets 0.100 mg ZEN/kg feed (100 ppb). In this study, the effects of ZEN in concentrations below (75 ppb) and above (290 ppb) EU recommendation on the level of some key markers involved in the oxidative and inflammatory response, as well as the mechanisms and signaling pathways through which ZEN could produce its toxicity, were monitored in the colon of weaned piglets. The exposure of the piglets to the lower concentration of ZEN (75 ppb) did not lead to changes in stress and inflammation markers or in the signaling pathways associated with these processes.

**Keywords:** zearalenone; piglets; weaning; inflammation; oxidative stress; signaling pathway

## 1. Introduction

Mycotoxins are some of the most common natural contaminants in food or feed involving not only health issues but also enormous economic losses [1]. Fusariotoxins are among the most known toxins produced by molds of the *Fusarium* genus and the largest group, which includes over 140 known metabolites. The most common fusariotoxins are zearalenone, deoxynivalenol, nivalenol, T-2, HT-2 toxins, and fumonisins [2]. Farm animals are often affected by mycotoxins, particularly the swine species due to the high cereal content in the diet and their native sensitivity [3].

Zearalenone (ZEN) is a fusariotoxin belonging to the xenoestrogen class due to its structural similarity to β-estradiol and its affinity for estrogen receptors [4]. It was demonstrated that ZEN causes reproductive disorders, including hormonal and reproductive disorders, especially in pigs. Moreover, ZEN affects also other physiological systems: digestive, immune, and nervous system, which are manifested by diarrhea, vomiting, reduced appetite, leukopenia, etc. [5–7].

According to the European Commission (Recommendation EC/2006/576), the guideline values for ZEN as allowed limit in feed are 0.100 mg ZEN/kg (100 ppb) for piglets and young sows and 0.250 mg ZEN/kg (250 ppb) for mature sows and pigs [8]. These values are only recommendations and more toxicological data issued from in vivo studies are necessary in order to establish a regulation for the concentration of zearalenone that can be admitted in feed material and compound feed for swine. Moreover, there are studies on

pigs that reported significant effects at a concentration of 100 ppb ZEN in the feed [9] and others in which this concentration had no statistically notable effects [10].

In order to bring new scientific data leading to the establishment of a guideline value for ZEN in young pigs, the aim of this study was to evaluate in vivo the effects of two concentrations of ZEN, one below the EC recommendation (75 ppb) and the other above the EC recommendation (290 ppb) on piglets immediately after weaning.

Weaning is a difficult period for piglets due to the switch from sow's milk to solid feed, environmental changes, and an undeveloped immune and digestive system which predispose piglets to pathogen infections and digestive disorders including diarrhea [11]. The quality of the feed is of great importance in this period for the development of digestive and defense systems [12].

During the post-weaning period (1–2 weeks) the gut, including the colon are the most vulnerable tissue [13]. Currently, many studies on the exposure of weaned piglets to ZEN are performed at the small intestine level. However, the large intestine takes up undigested feed; therefore, at the level of the colon, the absorption of several substances can take place, excluding water and electrolytes [14]. Also, the colon is a gut segment with high microbial activity (active substrate hydrolysis and fermentation). As shown by Richards et al. [15] colonic microbiota produced fermentation products with protective function stimulating the immune response. But, at weaning, several defaults in the intestinal barrier function were observed which could be a starting point for inflammation, as well as water and electrolyte imbalance [16]. During this period, an increase in secretory activity and permeability was observed in the colon [17]. Moreover, diarrhea frequently occurred during the weaning period and is often associated either with the shifts in microbiota (suppression of several beneficial lactic bacteria) after the transition to solid feed or with colonic inflammation caused by enterotoxigenic *Escherichia coli* or other pathogens [18]. These pathogens increased the inflammatory cytokines in the colon, contributing to colonic inflammation. Zearalenone can be an additional pro-inflammatory factor that increases colonic inflammation if the feed that the piglets eat during this period is contaminated with ZEN. Bauer et al. [19] reported that feed type and quality had an important influence on gut microbiota. Reddy et al. [7], analyzing the colon content of three groups of pigs fed diets contaminated with zearalenone (800 ppb) and deoxynivalenol (8000 ppb), reported that the toxin-contaminated diets significantly affected the colon microbiota, especially *Lactobacillus*. Similarly, a previous study of our team found also that ZEN (290 ppb in the diet) decreased the populations of *Lactobacillus* and *Bifidobacterium* [20]. However, the effects of ZEN on the pig colon microbiota, and inflammatory and stress response are not completely understood [7]. Thus, the effect on innate (toll like receptors), pro-inflammatory and anti-inflammatory immune response (cytokines), oxidative/antioxidant response (lipid, protein, nucleic acids, and antioxidant enzymes), and the underlying signaling pathways (NF-kB, KEAP1, HO1, NQO1, Nrf2 etc.) were investigated in the colon of weaned piglets exposed to low (75 ppb) and for comparison to a high dose (290 ppb) of ZEN contaminated diet.

## 2. Materials and Methods

### 2.1. Toxin Preparation

Zearalenone purchased from FERMENTEK (Jerusalem, Israel) was dissolved in dimethyl sulfoxide (DMSO). The obtained solution was diluted in water at a ratio of 1:7 and then added to the basal diet until final concentrations of 75 µg/kg feed and 290 µg/kg feed were obtained. Considering the occurrence of mycotoxins naturally in feed, the presence of aflatoxin B1 (AFB1), deoxynivalenol (DON), ochratoxin A (OTA), total fumonisins (FBs: B1, B2, B3), T2/HT-2, and ZEN were analyzed by ELISA, using the VERATOX (Neogene, Lansing, MI, USA) kit according to the manufacturer's instructions, with a detection limit between 0.1 and 200 ppb (0.5 ppb AFB1, 100 ppb DON; 1 ppb OTA, 0.2 ppm FBs, T2/HT-2 25 ppb, 5 ppb ZEN). The concentration of all investigated mycotoxins was below the detection limit. The diets were also screened for contamination with bacteria (total bacteria

count, *Escherichia coli*, *Salmonella*) and with fungus (total fungi count) and the contamination levels were found to be under the EU limits accepted for pigs.

## 2.2. Experimental Design

The experiment was performed in vivo on 18 crossbred weaned piglets (TOPIGS-40) hybrid [female Large White × Hybrid (Large White × Pietrain) × male Talent, mainly Duroc]. Hybrid piglets were weaned at 27 days of age with a body weight of $11.25 \pm 1.14$ kg, randomly assigned to three groups (6 animals/group), control group (Control) fed with uncontaminated feed and two experimental groups: group experimental 1 (ZEN 75) fed the control diet artificially contaminated with 75 µg ZEN/kg feed below the concentration recommended by the EC and group experimental 2 (ZEN290) fed the control diet artificially contaminated with 290 µg ZEN/kg feed above the EC recommendation. The hybrid piglets were weaned at 27 days of age with a body weight of $11.25 \pm 1.14$ kg. The ingredients of the basal diet were described by Grosu et al., 2023 [20]. Briefly, the diet contained corn (68.46%), soybean meal (19%), vegetal milk (5%), corn gluten (4%), l-Lysine (0.31%), Methionine (0.10%), $CaCO_3$, (1.57), Ca $(H_2PO_4)_2$, mineral–vitamin premix (1%), choline premix (0.10%), and phytase (0.01%). The diets were screened for other mycotoxin contamination, and the levels were found to be under the EU limits for pigs. The experiment lasted 30 days and the animals had free access to feed and water. At the end of the experiment, animals were killed (electrically stunned) and colon samples were collected perfused with ice-cold saline solution and stored at $-80\ °C$ until further analysis. The experiment complied with the EU Council Directive 98/58/EC and Romanian Law 206/2004 which regulates treating animals used for experimental purposes. The experimental design was approved by the Ethical Committee of the National Research-Development Institute for Animal Nutrition and Biology, Balotesti, Romania (Ethical Committee no. 41/2023).

## 2.3. Quantification of Gene Expression

The effects produced by the low and the higher concentration of ZEN on the gene expression of several inflammatory cytokines, such as interleukin1-beta (*IL-1β*), tumor necrosis factor-alpha (*TNF-α*), interleukin-8 (*IL-8*), interleukin-6 (*IL-6*), interleukin-4 (*IL-4*), interleukin-10 (*IL-10*), interferon-gamma (*IFN-γ*); antioxidant enzymes superoxide dismutase (*SOD*), glutathione peroxidase (*GPx*), and catalase (*CAT*); Toll-like receptors *TLR 2*, *TLR 4*, *TLR 5*, and *TLR 9*; signaling molecules *TGFβ2*, *MYD88*, *IRAK1*, *TRAF6*, *TAK1*, *AKT*, *JNK 1/2/3*, *ERK 1/2*, and *p38α*; and transcription factors nuclear factor kappa b (*NF-kB*), activator protein 1 (*AP-1*), and nuclear factor erythroid 2-related factor 2 (*Nrf-2*), involved in inflammation and oxidative stress pathways were analyzed. The RNA extraction and transcription into cDNA were performed using the protocol described by Marin et al. [5]. Subsequently, 10 ng of cDNA sample was mixed with 0.3 µM primers of the genes of interest and 10 µL of SYBR Green qPCR Master Mix (Life Technologies, Carlsbad, CA, USA), and brought to the final volume of 25 µL with RNAse water free. The reactions took place in several steps: 2 min at $50\ °C$, 10 min at $95\ °C$, followed by 40 cycles of 15 s at $95\ °C$ and 60 s at $60\ °C$, completed with an elongation of 10 min at $72\ °C$. Data obtained were normalized with Norm Finder (x93NormFinderx94) software, and two reference genes were selected from a total of 6 housekeeping genes analyzed. Nucleotide sequences of the primers used in the experiment are shown in Table 1. The calculation method used was $2^{(-\Delta\Delta Ct)}$, and results were expressed as Fold Change (FC) compared to the control group.

**Table 1.** Nucleotide sequences of the primers.

| Gene | Primer Sequence | Orientation | Tm (°C) | Amplicon Length (bp) |
|---|---|---|---|---|
| β-2 microglobulin | TTCTACCTTCTGGTCCACACTGA | 5′-3′ | 50 | 162 |
| | TCATCCAACCCAGATGCA | 3′-5′ | 54 | |
| GAPDH | ACTCACTCTTCTACCTTTGATGCT | 5′-3′ | 49 | 100 |
| | TGTTGCTGTAGCCAAATTCA | 3′-5′ | 56 | |
| Cyclophilin A | CCCACCGTCTTCTTCGACAT | 5′-3′ | 54 | 92 |
| | TCTGCTGTCTTTGGAACTTTGTCT | 3′-5′ | 55 | |
| β-actin | GGACTTCGAGCAGGAGATGG | 5′-3′ | 60 | 230 |
| | GCACCGTGTTTGCGTAGAGG | 3′-5′ | 62 | |
| HGPRT-1 | TGGAAAGAATGTCTTGATTGTTGAAG | 5′-3′ | 58.57 | 93 |
| | ATCTTTGGATTATGCTGCTTGACC | 3′-5′ | 59.66 | |
| RPL 32 | TGCTCTCAGACCCCTTGTGAAG | 5′-3′ | 61.93 | 106 |
| | TTTCCGCCAGTTCCGCTTA | 3′-5′ | 59.63 | |
| TNF-α | ACTGCACTTCGAGGTTATCGG | 5′-3′ | 60 | 118 |
| | GGCGACGGGCTTATCTGA | 3′- 5′ | 60 | |
| IL-8 | GCTCTCTGTGAGGCTGCAGTTC | 5′-3′ | 58 | 79 |
| | AAGGTGTGGAATGCGTATTTATGC | 3′-5′ | 54 | |
| IL-6 | GGCAAAAGGGAAAGAATCCAG | 5′-3′ | 57 | 87 |
| | CGTTCTGTGACTGCAGCTTATCC | 3′-5′ | 61 | |
| IL-1β | ATGCTGAAGGCTCTCCACCTC | 5′-3′ | 62 | 89 |
| | TTGTTGCTATCATCTCCTTGCAC | 3′-5′ | 59 | |
| IFNγ | TGGTAGCTCTGGGAAACTGAATG | 5′-3′ | 54 | 79 |
| | GGCTTTGCGCTGGATCTG | 3′-5′ | 55 | |
| IL-4 | CAACCCTGGTCTGCTTACTG | 5′-3′ | 52 | 173 |
| | CTTCTCCGTCGTGTTCTCTG | 3′-5′ | 52 | |
| IL-10 | GGCCCAGTGAAGAGTTTCTTTC | 5′-3′ | 54 | 51 |
| | CAACAAGTCGCCCATCTGGT | 3′-5′ | 55 | |
| NF-kB | CGAGAGGAGCACGGATACCA | 5′-3′ | 55 | 62 |
| | GCCCCGTGTAGCCATTGA | 3′-5′ | 54 | |
| CAT | CTTGGAACATTGTACCCGCT | 5′-3′ | 62 | 241 |
| | GTCCAGAAGAGCCTGAATGC | 3′-5′ | 62 | |
| GPx | GGAGATCCTGAATTGCCTCAAG | 5′-3′ | 50 | 62 |
| | GCATGAAGTTGGGCTCGAA | 3′-5′ | 58 | |
| SOD | GAGACCTGGGCAATGTGACT | 5′-3′ | 62 | 139 |
| | CTGCCCAAGTCATCTGGTTT | 3′-5′ | 60 | |
| Nrf2 | CCCATTCACAAAAGACAAACATTC | 5′-3′ | 57 | 72 |
| | GCTTTTGCCCTTAGCTCATCTC | 3′-5′ | 59 | |
| AP1 | CCCAAGATCCTGAAGCAGAG | 5′-3′ | 62 | 136 |
| | GATGTGCCCGTTACTGGACT | 3′-5′ | 62 | |

**Table 1.** *Cont.*

| Gene | Primer Sequence | Orientation | Tm (°C) | Amplicon Length (bp) |
|---|---|---|---|---|
| p38α | TGCAAGGTCTCTGGAGGAAT | 5′-3′ | 52 | 109 |
| | CTGAACGTGGTCATCCGTAA | 3′-5′ | 52 | |
| TGFβ2 | CGATGATGATGTTGATGATGG | 5′-3′ | 55 | 69 |
| | GCAAGGCTTTCTTGTATTTTCTTG | 3′-5′ | 58 | |
| TAK1 | TGCCCAAACTCCAAAGAATC | 5′-3′ | 56 | 151 |
| | TTTGCTGGTCCTTTTCATCC | 3′-5′ | 56 | |
| ERK1 | CTACCTGGACCAGCTCAACC | 5′-3′ | 60 | 85 |
| | CACTGTGATCCGTTTGTTGG | 3′-5′ | 60 | |
| ERK2 | TGACATTCAACCCTCACAAGA | 5′-3′ | 42.86 | 198 |
| | ATCTGTATCCTGGCTGGAATC | 3′-5′ | 47.62 | |
| JNK1 | TGCTTTGTGGAATCAAGCAC | 5′-3′ | 51 | 60 |
| | TGGGCTTTAAGTCCCGATG | 3′-5′ | 51 | |
| JNK2 | TATTATCGGGCACCAGAAGTC | 5′-3′ | 51 | 97 |
| | AACCTTTCACCAGCTCTCTCA | 3′- 5′ | 53 | |
| JNK 3 | TGCCTATGACGCTGTTCTTG | 5′-3′ | 58.27 | 180 |
| | TGAAACTCCTCCAGCGTCTT | 3′-5′ | 58.95 | |
| HO-1 | ATGTGAATGCAACCCTGTGA | 5′-3′ | 57.71 | 89 |
| | GGAAGCCAGTCAAGAGACCA | 3′-5′ | 59.31 | |
| NQO1 | GTATCCTGCCGAGACTGCTC | 5′-3′ | 59.97 | 134 |
| | TAGCAGGGACTCCAAACCAC | 3′-5′ | 59.31 | |
| KEAP | ACGACGTGGAGACAGAAACGT | 5′-3′ | 61.94 | 56 |
| | GCTTCGCCGATGCTTCA | 3′-5′ | 58.07 | |
| Akt | AAGGCCACGGGCCGCTACTA | 5′-3′ | 65.94 | 100 |
| | GGAGGACGCGGTTCTCCGT | 3′-5′ | 64.2 | |
| MyD88 | GCAGCTGGAACAGACCAACT | 5′-3′ | 60 | 66 |
| | GTGCCAGGCAGGACATCT | 3′-5′ | 59 | |
| MD-2 | CCTTGTTTTCTTCCATATTTACTG | 5′-3′ | 54 | 63 |
| | CATCAGAGGAATTGCAGATCCA | 3′- 5′ | 58 | |
| IRAK1 | CAAGGCAGGTCAGGTTTCGT | 5′-3′ | 55 | 115 |
| | TTCGTGGGGCGTGTAGTGT | 3′-5′ | 58 | |
| TRAF-6 | CAAGAGAATACCCAGTCGCACA | 5′-3′ | 50 | 122 |
| | ATCCGAGACAAAGGGGAAGAA | 3′-5′ | 48 | |
| TLR2 | TCACTTGTCTAACTTATCATCCTCTTG | 5′-3′ | 59 | 162 |
| | TCAGCGAAGGTGTCATTATTGC | 3′-5′ | 59 | |
| TLR4 | GCCATCGCTGCTAACATCATC | 5′-3′ | 60 | 108 |
| | CTCATACTCAAAGATACACCATCGG | 3′-5′ | 59 | |
| TLR5 | CCTTCCTGCTTCTTTGATGG | 5′-3′ | 56 | 124 |
| | CTGTGACCGTCCTGATGTAG | 3′-5′ | 57 | |
| TLR9 | CACGACAGCCGAATAGCAC | 5′-3′ | 59 | 121 |
| | GGGAACAGGGAGCAGAGC | 3′-5′ | 60 | |

Results for gene expression were validated through western blot and enzyme-linked immunosorbent assay (ELISA) techniques for the following proteins: Nrf-2 and NF-kB, as well as IL-1β, TNF-α, IL-8, and IFN-γ.

### 2.4. Quantification of Protein Expression by Western Blot

The phosphorylated form of protein expression of two important nuclear transcription factors Nrf-2 and NF-kB, key molecules which regulate inflammation and oxidative stress pathways was determined by Western Blot. Protein concentration in colon sample lysates in RIPA buffer was quantified using a specific kit (Pierce BCA Protein Assay Kit, Thermo Fischer Scientific, Waltham, MA, USA). The protein (30 μg) was separated by 10% SDS-page electrophoresis, transferred to the nitrocellulose membrane, blocked, and incubated with specific primary and secondary antibodies as described by Pistol et al. [21] using the same buffers, antibodies, and reagents, as well as the time of incubation. The immunoblotting images were developed using a MicroChemi Imager (DNR Bio-Imaging Systems Ltd., Neve Yamin, Israel) and the protein expression level was evaluated using GelQuant software (DNR Bio-Imaging Systems Ltd., Neve Yamin, Israel). The results of target protein expression were reported to β-actin housekeeping protein expression and reported as arbitrary units.

### 2.5. Quantification of Protein Concentration (ELISA)

Protein concentration for several pro-inflammatory cytokines, IL-1β, TNF-α, IL-8, and IFN-γ, was determined by using the ELISA technique and specific kits provided by R&D Systems (Minneapolis, MN, USA) according to the manufacturer's instructions as described by Pistol et al. [21]. At the end of the reaction, the optical density was measured using a plate reader (SUNRISE TECAN, Grödig, Austria), and the protein concentrations were calculated by reference to a standard curve provided by the kit.

### 2.6. Assessment of Oxidative Stress

Oxidative stress was evaluated at three levels: lipid (TBARS), protein (protein carbonyl), and nucleic acids (DPA assay).

### 2.6.1. Lipid Peroxidation Measurement (Thiobarbituric Acid Reactive Substances-TBARS)

The lipid oxidation was determined from a 0.2-g colon sample homogenized in PBS (phosphate buffered saline), and thiobarbituric acid reactive substances (TBARS) were measured according to the protocol already described by Marin et al. [22]. Results were expressed as nmol/mL.

### 2.6.2. Protein Oxidation Measurement (Protein Carbonyl Analysis)

The protein oxidation was evaluated based on the detection of the reaction product between 2,4-dinitrophenyl hydrazine with protein carbonyls, known as the protein hydrazone using spectrophotometric methods. The protein concentrations from colon samples were determined by using the Pierce BCA Protein Assay Kit (ThermoFisher Scientific, Rockford, IL, USA). The absorbances were determined at a wavelength of 370 nm using a microplate reader (Tecan, Sunrise, Vienna, Austria), the results being expressed in nmol/mg of carbonyl content.

### 2.6.3. DNA Fragmentation Using Diphenylamine (DPA)

To quantify DNA fragmentation in the colon samples, the DPA test was performed according to the method described by Ben Salah-Abbes et al. [23]. The tissue (0.05 g) was homogenized in 0.5 mL of lysis buffer (10 mM tris-HCl pH 8, 0.2% triton X-100, 1 mM EDTA) and centrifuged for 20 min at 4 °C and $10,000 \times g$. The supernatant (S) was collected, the pellet (P) was resuspended in 0.5 mL of lysis buffer, and 0.5 mL of 25% trichloroacetic acid was added to both the supernatant and the pellet. The samples were incubated for 24 h at 4 °C. After incubation, the samples were centrifuged, the supernatant discarded,

and 80 μL of 5% TCA was added to the obtained precipitates and left for 20 min at 83 °C. The diphenylamine indicator (DPAi) was prepared using two solutions, the first with 1.5 g diphenylamine, 1.5 mL sulfuric acid, and 100 mL glacial acetic acid, and the second solution of 16 mg/mL acetic aldehyde. A volume of 160 μL of DPAi was added to each sample which was stored at room temperature for 24 h. At the end, the optical density (OD) was read at 600 nm using a microplate reader (Tecan, Sunrise, Vienna, Austria). DNA fragmentation was calculated according to the formula:

$$\% \text{ Fragmented DNA} = \frac{\text{OD(S)}}{\text{OD(S)} + \text{OD(P)}} \times 100$$

*2.7. Assessment of Antioxidant Response*

2.7.1. Determination of Total Antioxidant Status (TAC)

The Total Antioxidant Status was determined using a spectrophotometric method described by Taranu et al. [24]. The absorption of 20-azinobis-[3-ethylbenzothiazoline-6-sulfonic acid cation (ABTS+)] was measured from colon samples at the wavelength of 732 nm, the results being expressed in mmol/L Trolox equivalents.

2.7.2. Determination of Antioxidant Enzyme Activity

The antioxidant enzymes Superoxide Dismutase (SOD) and Catalase (CAT) activity was assessed using Cayman kits (Cayman Chemical, Ann Arbor, MI, USA) according to the instructions provided by the manufacturer as described by Chedea et al. [25]. The absorbances were read using the Tecan microplate reader (Sunrise, Vienna, Austria) at a wavelength of 450 nm.

*2.8. Statistical Analyses*

The results are represented graphically by mean ± standard error of the mean (SEM). A one-way ANOVA test was performed using GraphPad Prism (9.3.0) software, followed by Fisher's exact test, and the differences between the experimental groups were considered significant at a $p$-value < 0.05; $p$ values between 0.05 and 0.1 being considered a trend. The statistical significance was marked graphically as follows: ns—not significant ($p$-value $\geq$ 0.05), *—significant ($0.05 \geq p$-value > 0.01), **—very significant ($0.01 \geq p$-value > 0.001) and ***—extremely significant ($0.001 \geq p$-value > 0.0001).

**3. Results**

*3.1. Effect of ZEN on Innate Immunity*

Analyzing the gene expression of some TLRs involved in the innate immune response (Figure 1), it was observed that exposure to a higher concentration of ZEN leads to a significant increase in the gene expression level of TLR 4 (60% increased, $p$ = 0.0236), while at the level of TLR2, TLR5, and TLR9, no change is observed. Moreover, along with the significant increase in TLR4 gene expression, the level of MYD88 gene expression increased significantly (48% increased, $p$ = 0.0399), in the colon of piglets exposed to 290 ppb ZEN as compared with the control, suggesting that the TLR4/MYD88 signaling pathway could be involved in ZEN toxicity.

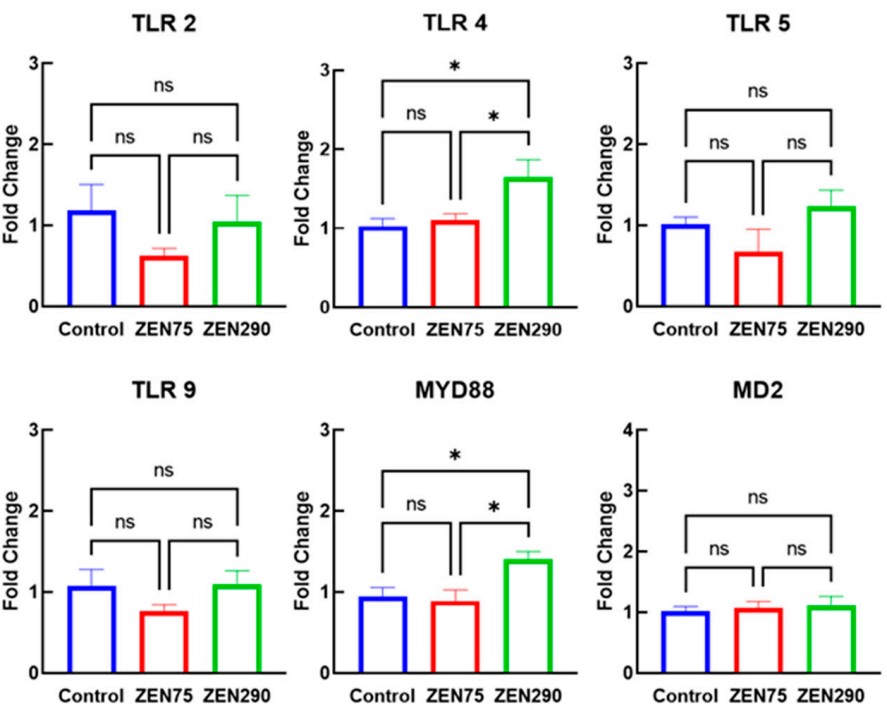

**Figure 1.** Effect of ZEN on gene expression of toll-like receptors TLR2, TLR 4, TLR5, and TLR9, and MYD88 and MD2 receptors in the colon of weaned piglets exposed to a concentration lower (75 ppb) or above (290 ppb) EC recommendation in feed. The statistical significance was marked graphically as follows: ns—not significant ($p$-value $\geq$ 0.05), *—significant (0.05 $\geq$ $p$-value > 0.01).

### 3.2. Effect of ZEN on Oxidative Response

Regarding oxidative stress, at all three analyzed levels of lipid (TBARS), protein (protein carbonyl), and nucleic acids (DNA fragmentation), neither of the two concentrations of ZEN induced significant changes. However, in the case of protein oxidation, an increasing trend ($p$ = 0.069) is observed for ZEN 290 (Figure 2).

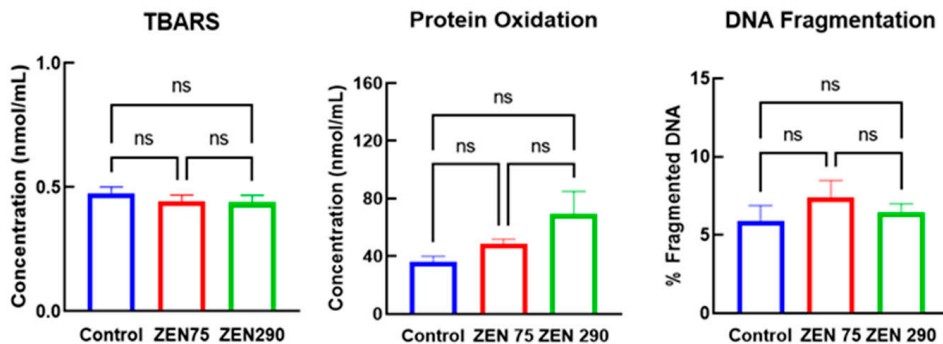

**Figure 2.** Effect of ZEN on lipid peroxidation (TBARS), protein oxidation, and DNA fragmentation in the colon of weaned piglets exposed to a concentration lower (75 ppb) or above (290 ppb) EC recommendation in feed. The statistical significance was marked graphically as follows: ns—not significant ($p$-value $\geq$ 0.05).

### 3.3. Effect of ZEN on Antioxidant Response

Regarding the total antioxidant status, a significant decrease was recorded in the case of ZEN 75 ($p$ = 0.025) and ZEN 290 ($p$ = 0.006). Knowing that the imbalance between antioxidants and oxidants plays an important role in oxidative status, the decrease in TAC indicates potential oxidative stress induced by ZEN exposure (Figure 3). Interestingly, the results on genes encoding for antioxidant enzymes CAT, SOD, and GPx showed that

both concentrations of ZEN below and above the EC recommendation did not significantly modify their expression in the colon of piglets fed the contaminated diet. Furthermore, the results obtained by qPCR were confirmed by enzymatic activity of CAT and SOD; ZEN did not induce any significant changes.

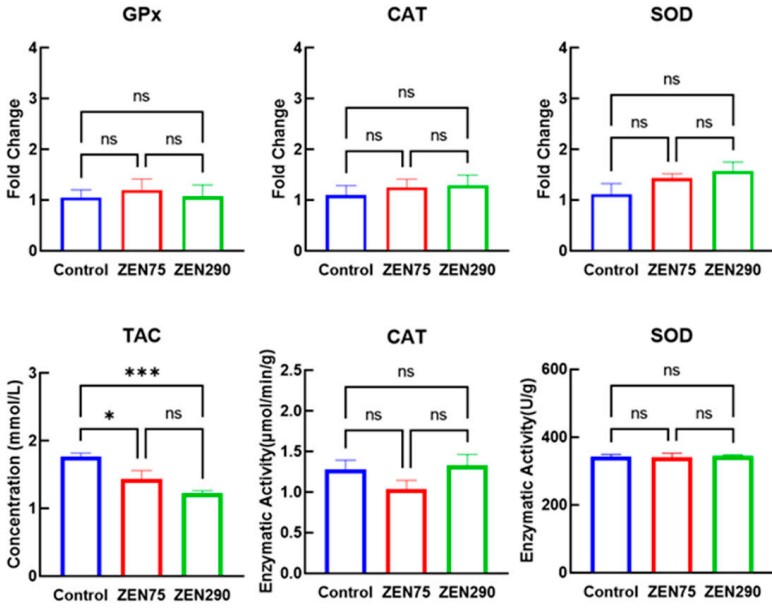

**Figure 3.** Effect of ZEN on total antioxidant capacity (TAC) and antioxidant enzymes CAT, SOD, and GPx in the colon of weaned piglets exposed to a concentration lower (75 ppb) or above (290 ppb) EC recommendation in feed. The statistical significance was marked graphically as follows: ns—not significant (*p*-value ≥ 0.05), *—significant (0.05 ≥ *p*-value > 0.01) and ***—extremely significant (0.001 ≥ *p*-value > 0.0001).

Analyzing the gene (Figure 4a) and protein expression (Figure 4b) of the Nrf2 signaling molecule responsible for the activation of antioxidant enzymes response, we did not observe a significant effect caused by any of the analyzed concentrations; however, in the case of ZEN 75, an increasing trend was observed in Nrf2 protein expression (*p* = 0.086).

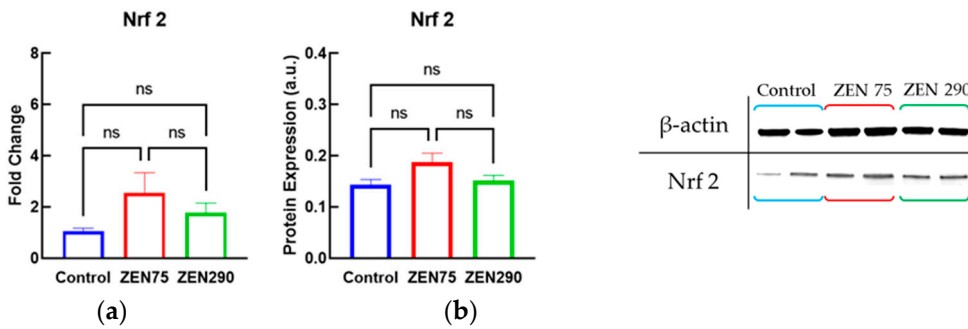

**Figure 4.** Effect of ZEN on (**a**) gene expression and (**b**) protein expression (histogram and protein detection image) of Nrf2 nuclear receptor in the colon of weaned piglets exposed to a concentration lower (75 ppb) or above (290 ppb) EC recommendation in feed. The statistical significance was marked graphically as follows: ns—not significant (*p*-value ≥ 0.05).

The gene expression of KEAP1, HO1, and NQO1 was also analyzed, knowing that they are key markers on the main regulatory KEAP1/Nrf2 pathway that triggers the intracellular defense against oxidative stress [26]. As in the case of Nrf2, no significant changes were observed regarding the gene expression levels of KEAP1, HO1, and NQO1 (Figure 5).

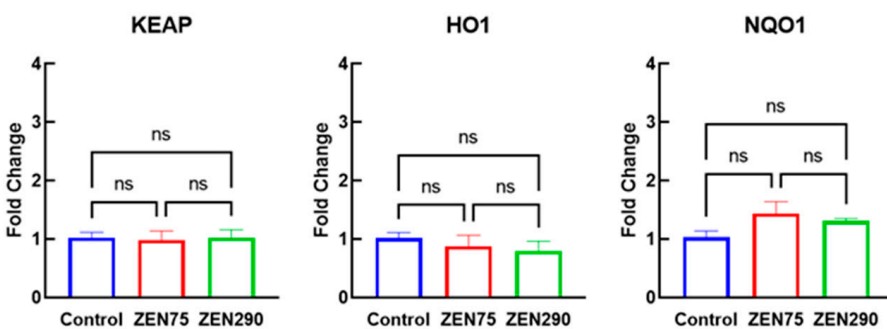

**Figure 5.** Effect of ZEN on gene expression of oxidative stress markers KEAP1, HO1, and NQO1 in the colon of weaned piglets exposed to a concentration lower (75 ppb) or above (290 ppb) EC recommendation in feed. The statistical significance was marked graphically as follows: ns—not significant ($p$-value $\geq 0.05$).

### 3.4. Effect of ZEN on Pro-Inflammatory Response

The results obtained from the qPCR analysis showed that at the colon level, neither the concentration of ZEN 75 ppb, nor that of 290 ppb significantly affected the inflammatory response evaluated through the gene expression of the powerful inflammatory cytokines IL-1β, TNF-α, IL-8, IFN-γ, and IL-6 compared to the control group (Figure 6). Although a higher level of expression of these genes was recorded under 290 ppb of ZEN, the differences against the control group were insignificant.

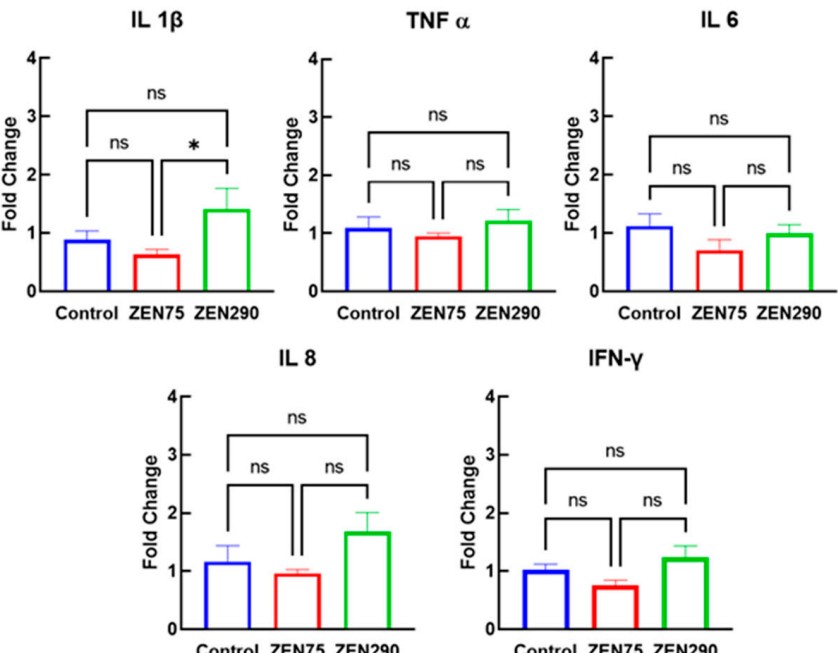

**Figure 6.** Effect of ZEN on gene expression of pro-inflammatory cytokines IL-1β, TNF α, IL 6, IL 8, and IFN-γ in the colon of weaned piglets exposed to a concentration lower (75 ppb) or above (290 ppb) EC recommendation in feed. The statistical significance was marked graphically as follows: ns—not significant ($p$-value $\geq 0.05$), *—significant ($0.05 \geq p$-value $> 0.01$).

To validate the qPCR results obtained for the investigated pro-inflammatory markers, further analyses on the effect of the two concentrations of ZEN on their proteins were performed. The protein concentration determined by the ELISA method confirms the results obtained by qPCR for cytokines IL-1β, TNF-α, IL-8, and IFN-γ. No significant differences in protein concentration between the control and the two treatments were observed for the pro-inflammatory cytokines IL-1β, TNF-α, IL-8, and IFN-γ (Figure 7).

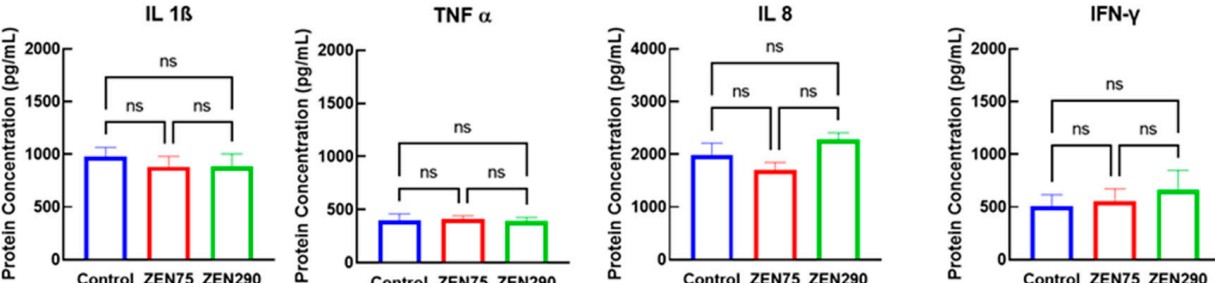

**Figure 7.** Effect of ZEN on the protein concentration of pro-inflammatory cytokines IL-1β, TNF α, IL 8, and IFN-γ in the colon of weaned piglets exposed to a concentration lower (75 ppb) or above (290 ppb) EC recommendation in feed. The statistical significance was marked graphically as follows: ns—not significant (*p*-value ≥ 0.05).

### 3.5. Effect of ZEN on Anti-Inflammatory Response

Furthermore, the effects of exposure to either ZEN 75 ppb or ZEN 290 ppb were investigated on the anti-inflammatory response expression in the colon through the assessment of cytokines IL-4 and IL-10 gene expression. Similarly, with the results obtained for pro-inflammatory cytokines, the gene expression level coding for the two interleukins IL-4 and IL-10 did not undergo statistically significant changes (Figure 8).

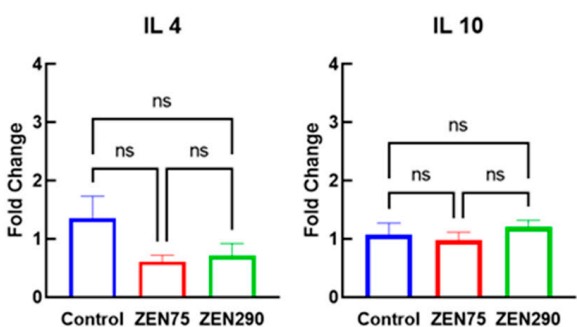

**Figure 8.** Effect of ZEN on gene expression of anti-inflammatory cytokines IL 4 and IL 10 in the colon of weaned piglets. The statistical significance was marked graphically as follows: ns—not significant (*p*-value ≥ 0.05).

### 3.6. Effect of ZEN on Molecules Involved in Signaling Pathways

The gene and protein expression of the NF-kB nuclear receptor, a key marker of the signaling pathway involved in inflammation and oxidative stress, did not undergo significant changes in 75 ppb ZEN group compared to the control, while in the case of 290 ppb ZEN group, a significant increase was observed for both genes (43% increase, *p* = 0.028, Figure 9a) and protein expression (24% increase, *p* = 0.0013, Figure 9b) of NF-kB when compared to control.

Figure 10 shows the gene expression of MAPKs analyzed in the colon of control and ZEN exposed piglets. A low concentration of ZEN did not induce significant changes compared to the control. However, exposure to ZEN 290 produces significant changes, increasing the gene expression level of *ERK1* (90% increase, *p* = 0.0196), *ERK2* (68% increase, *p* = 0.0261), *p38α* (72% increase, *p* = 0.0131), *JNK2* (67% increase, *p* = 0.0062), and *JNK3* (74% increase, *p* = 0.0511). Moreover, exposure to 290 ppb ZEN leads to a decrease in the gene expression level of the *NLRP6* inflammasome, which plays an important role in maintaining homeostasis at the gut level, while at the level of the *NLRP3* inflammasome, no change is observed due to exposure to ZEN.

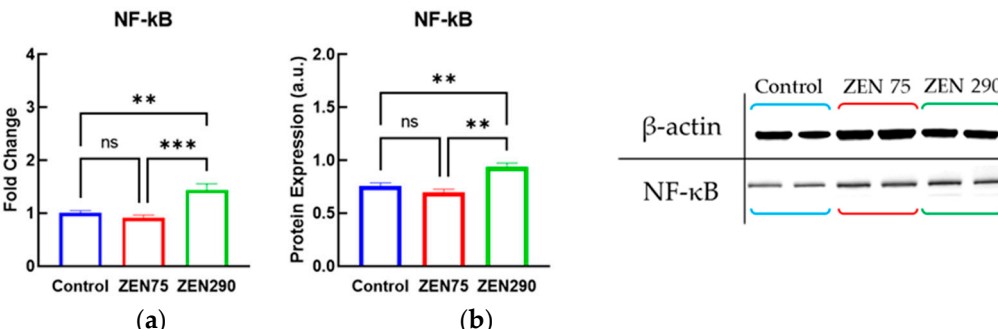

**Figure 9.** Effect of ZEN on (**a**) gene expression and (**b**) protein expression (histogram and protein detection image) of NF-kB nuclear receptor in the colon of weaned piglets exposed to a concentration lower (75 ppb) or above (290 ppb) EC recommendation in feed. The statistical significance was marked graphically as follows: ns—not significant (*p*-value ≥ 0.05), **—very significant (0.01 ≥ *p*-value > 0.001) and ***—extremely significant (0.001 ≥ *p*-value > 0.0001).

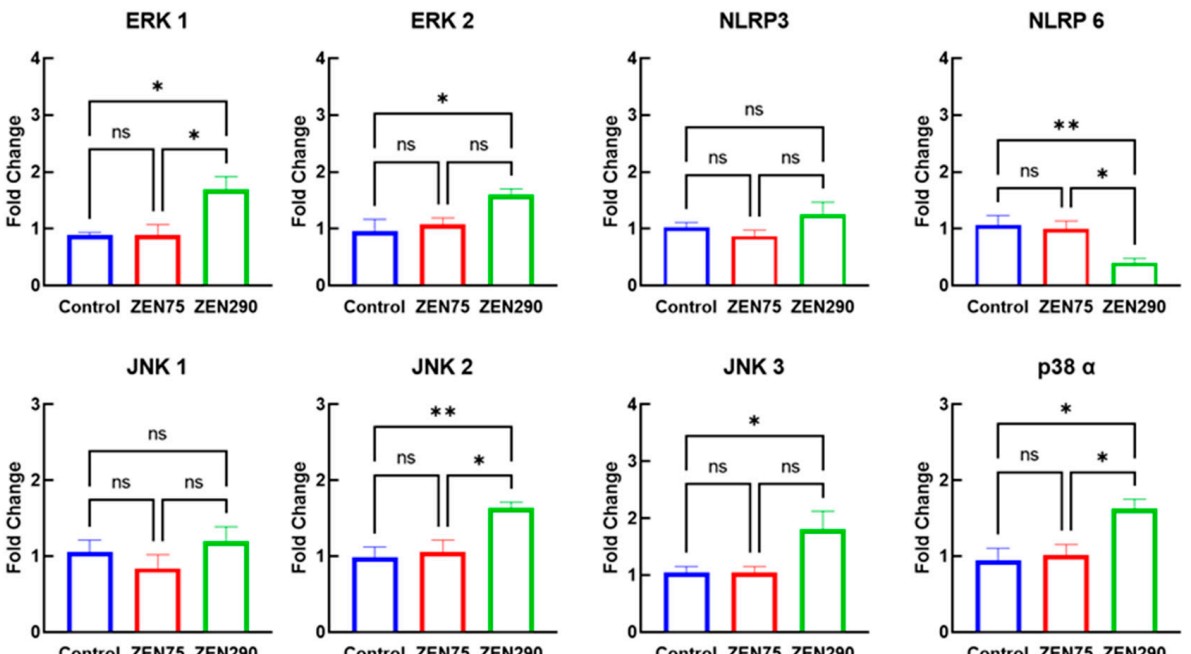

**Figure 10.** Effect of ZEN on gene expression of MAPKs ERK 1, ERK 2, JNK 1, JNK 2, JNK 3, and p38α, and inflammasomes NLRP 3 and NLRP 6 in the colon of weaned piglets. The statistical significance was marked graphically as follows: ns—not significant (*p*-value ≥ 0.05), *—significant (0.05 ≥ *p*-value > 0.01), **—very significant (0.01 ≥ *p*-value > 0.001).

Similar results were obtained in the case of the gene expression of other signaling molecules (Figure 11). As in the case of MAPKs, ZEN 75 did not induce significant changes, while ZEN 290 induced a significant increase compared with control in the gene expression level of IRAK1 (76%), TRAF 6 (67%), TAK 1 (64%), TGFβ2 (103%), AP1 (88%), and AKT (81%).

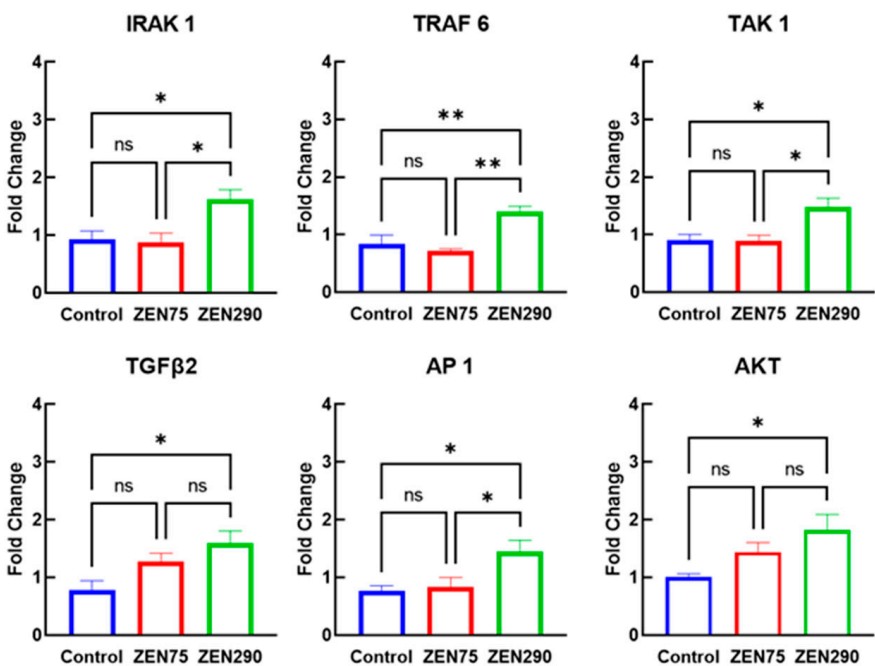

**Figure 11.** Effect of ZEN on gene expression of signaling molecules IRAK 1, TRAF 6, TAK 1, TGFβ2, AP1, and AKT in the colon of weaned piglets. The statistical significance was marked graphically as follows: ns—not significant ($p$-value ≥ 0.05), *—significant ($0.05 ≥ p$-value > 0.01), **—very significant ($0.01 ≥ p$-value > 0.001).

## 4. Discussions

Zearalenone is a fusariotoxin that frequently contaminates cereal crops and products derived from them. Its chemical structure not only gives it stability and temperature resistance but ZEN is also considered a xenoestrogenic compound due to its similarity to 17β-estradiol [4,27,28]. Considering the affinity for estrogen receptors, the main target of ZEN is the reproductive system, but toxic effects have been observed in the nervous, hepatic, and digestive systems [23]. Due to this toxicity, a regulation concerning the maximum allowed amount of ZEN in feed and food is necessary. Until now, at the European level, there are regulations only for grain and bakery products intended for humans (European Commission Regulation 1881/2006), with animal feed being only a recommendation (European Commission Recommendation 576/2006). For pigs, according to this document, the maximum level is 0.100 mgZEN/kg (100 ppb) feed for piglets, while for mature pigs and sows the level is 0.250 mg ZEN/kg feed (250 ppb) [29]. Considering the economic importance of pigs, and the health problems that ZEN-contaminated feed consumption can have on pigs, supplementary studies are necessary to establish if the EC recommendation might be taken as a norm (regulation) for the presence of ZEN in pig feed.

As the main route of exposure of piglets to ZEN is the oral one, its effects at the intestinal level are of major importance. Moreover, the intestine, colon included is one of the most sensitive organs for piglets during weaning. After the transition to solid feed diarrhea frequently occurred being associated either with the shifts in microbiota (suppression of several beneficial lactic bacteria) or with colonic inflammation caused by enterotoxigenic *Escherichia coli* or other pathogens (13). These pathogens increased the inflammatory cytokines in the colon contributing to colonic inflammation. Feed contaminated with mycotoxins could also aggravate the health of animals. That is why, in the present study, we investigated if a concentration of ZEN lower than the limit allowed by EC recommendation in the feed for piglets might produce toxic effects. A higher concentration of ZEN than the EC recommendation was used in the study for comparison. For this purpose, several key markers of innate and adaptative immune response were analyzed.

The innate immune response is responsible for recognizing and immediately combating foreign microorganisms, any disturbance of this system is directly correlated with inflammation [30]. Toll-like receptors, glycoproteins present on the cell surface or in the intracellular vesicles, play a very important role in recognizing and binding pathogens; they activate various defense mechanisms of the body, such as the synthesis of cytokines and chemokines [31,32]. No effect of feed with ZEN 75 ppb on TLRs was observed in our study. Only the exposure to the higher concentration of ZEN (290 ppb) leads to a significant increase in the gene expression level of *TLR 4* without modifying other TLRs such as *TLR 2*, *TLR 5*, and *TLR 9* (Figure 1). Studies on pregnant rats have shown that ZEN can induce oxidative stress and inflammatory response by increasing the level of inflammatory cytokines mediated by TLR4 [33]. Also, studies on cell intestinal porcine line IPEC 1 in vitro showed that combined exposure to ZEN and *Escherichia coli* lead to the upregulation of *TLR 2*, *TLR 3*, *TLR 4*, *TLR 6*, and *TLR 10* [34].

The activation of TLRs leads to the recruitment of adapters molecules such as MyD88, IRAK, or TRAF 6. The activation of IRAK or MYD88-dependent signaling pathway triggers complex signaling cascades mediated by MAPKs (p38$\alpha$, ERK1/2, JNK1/2/3), the process leading to the activation of transcription factors such as AP-1 and NF-kB, key modulatory factors of cytokine production [35]. In the case of co-exposure of Zebra fish to ZEN (200, 400, 800 $\mu$g/L) and DON (4000 $\mu$g/L), an increase in the expression of *TLR4*, *MYD88*, and *NF-kB* was also observed, leading to an increase in the level of ROS and antioxidant enzymes (CAT, SOD, and GPx). Although there are in vitro and in vivo studies related to TLR4-mediated ZEN toxicity, very few are carried out on pigs or pig cells. As mentioned in our study, only ZEN at 290 ppb produces an increase in the gene expression level of *TLR 4*. Moreover, in the case of ZEN290, the gene expression of *MYD88* is also increased, suggesting a potential mechanism mediated through the TLR 4-MYD88 signaling pathway.

Thus, oxidative stress is produced by the accumulation of reactive oxygen species (ROS) that have toxic effects, causing the oxidation of proteins, nucleic acids, and lipid membranes which can be correlated with TLR4-mediated pathways [36]. A decrease in the level of antioxidant enzymes was observed in the small intestine. Studies carried out on gilts showed that their exposure to a very high concentration of ZEN (1040 ppb) led to a significant decrease in the antioxidant enzymes CAT, SOD, and GPx activity in the ileum, duodenum, and jejunum of piglets [37]. It is known that ZEN can induce oxidative stress at the level of several other organs. Tests were performed in vitro on porcine Granulosa cells with induction of oxidative stress by low doses of ZEN (15 $\mu$M, 30 $\mu$M) while the mRNA level of the antioxidant enzymes SOD, CAT, and GPx decreased [38]. Studies carried out on weaned piglets show that exposure to ZEN (1.1 to 3.2 mg/kg feed) led to a decrease in the activity of the antioxidant enzymes SOD and GPx [39], enzymes that counteract oxidative stress. Similar effects were also observed in the spleen of weaned piglets that received feed contaminated with ZEN at 316 ppb, registering a decrease in the gene expression level of *SOD*. However, at the level of other organs, such as the liver, a totally opposite effect was observed, with the expression of CAT and GPx being increased [40]. In our study, at the level of the colon, a very important organ during the weaning period, neither ZEN 75 ppb nor ZEN 290 ppb produced oxidation at the protein, lipid, or nucleic level. Also, the level of gene expression of antioxidant enzymes was not significantly modified either by ZEN 75 ppb or by ZEN 290 ppb. However, this concentration produced a decrease in the total antioxidant capacity and a tendency to increase protein oxidation.

Oxidative stress is very closely associated with the inflammatory response, a complex process of defending the organism against toxins, viruses, bacteria, and other traumatic factors [41], mediated by proteins such as cytokines, bacterial lipopolysaccharides, and other chemical mediators [42]. Although cytokine secretion is generally specific to immune system cells, intestinal epithelial cells can also produce these mediators of the inflammatory response. The cytokines that can be synthesized by intestinal epithelial cells are IL 1, IL 10, IL 15, and TGF$\beta$, but other cytokines such as TNF $\alpha$, CCL20, IL 1$\beta$, IL 6, and IL8 are also expressed. Recent in vitro and in vivo studies show a dual character of ZEN, which can be

both pro and anti-inflammatory depending on the tissue type, concentration, and exposure time [27]. In pigs, it has been observed that the gene expression of certain pro-inflammatory cytokines increases in the case of bacterial infections [43]. During the weaning period, a transitory period of inflammation was observed in the intestine, in which an increase in the level of TNF α, IL β, and IL 6 is produced due to the change in diet [44]. Similarly, the increase in gene expression of pro-inflammatory cytokines IL-1β, TNFα, IL-6, IL-8, and IFN-γ was observed in young sows' colon, kidneys, and pancreas exposed to ZEN 100 ppb [10]. Also, a pro-inflammatory activity was reported in the case of daily exposure of weaned gilts to oral pills containing 5, 10, and 15 μg ZEN/b.w. where an increase in the level of pro-inflammatory cytokines IFN-γ, IL-1α, IL-1β, IL-2, IL-6, and TNFα was observed in the ileum after 14, 24, and 48 days of exposure.

In vitro studies on the porcine cell line IPEC 1 showed that ZEN in concentrations of 10–100 μM induced an increase in the synthesis of IL 8 and IL 10 cytokines after 24 h [45]. By contrast, at the systemic level an opposite effect was observed when pig PBMCs (peripheral blood mononuclear cell) were exposed 48 h to 5 and 10 μM ZEN, a decrease in IL 8 synthesis being registered [46].

Moreover, the exposure of pregnant sows to ZEN produced effects not only at the maternal level, but also affected the immunity of the newborn piglets. A study by Liu et al. [47] showed that an experimental diet with 270 μg ZEN/kg feed produced inflammation in the intestines of sows, but also increased the synthesis of pro-inflammatory cytokines TNF-α, IL-1α, IL-1β, and IL-6 in the jejunum of their piglets. Transmission of the inflammatory effect from mother to piglets can also occur during lactation; studies have shown that feeding piglets with milk from sows exposed to 300 ppb ZEN led to intestinal inflammation [48]. While studies show that there are inflammatory effects at the intestinal level when pigs are exposed to ZEN, the concentrations used within them are higher than those recommended by the European Commission of 100 ppb ZEN/kg feed for piglets and 250 ppb ZEN/kg feed for pigs and sows. The results obtained from this study show that the concentrations close to the recommended limit did not produce any significant changes, the exposure of piglets to 75 ppb and 290 ppb ZEN did not lead to major changes in the level of gene expression or protein concentrations of some inflammatory markers such as IL-1β, TNFα, IL-6, IL-8, IFN-γ, IL-4, and IL-10.

Taking into account the obtained results, we analyzed further if the mechanisms involved by ZEN in inflammation and oxidative stress are affected. As said before, one of the key factors that could be involved is the nuclear transcriptional factor NF-kB, which modulates a multitude of processes including oxidative stress, inflammation, innate immune response, cell proliferation, or apoptosis. NF-kB has an antagonistic activity with Nrf2; in conditions of oxidative stress the activity of NF-kB increases, while that of Nrf2 decreases, leading to an increase in pro-inflammatory cytokine synthesis. The NF-kB signaling pathway can also be inhibited, influencing protein oxidation and lipid peroxidation [49]. The activity of NF-kB can be mediated also by transforming growth factor-beta 2 (TGFβ-2), affecting the activity of the AP-1 transcriptional complex [50]. Concerning the intestinal epithelium, in vitro studies on the IPEC-J2 cell line showed that ZEN (6 μg/mL or 8 μg/mL) led to an increase in p38α gene expression, without inducing effects at the level of JNK1/2 and ERK1/2 [51]. In the case of the exposure of piglets to 316 ppb ZEN, a tendency to decrease the gene expression of NF-kB was observed in the spleen, a key organ in immunity, while other transcriptional regulators AP-1 registered a slight increase [52]. Another study shows similar effects, a decrease in the level of NF-kB in the liver of weaned piglets when experimental diet was contaminated with 250 ppb ZEN [53]. Moreover, the same authors reported a decrease in the gene expression level of TAK1/p38α, suggesting the involvement of MAPKs in the signaling pathways of ZEN. The results obtained in this work also suggest that ZEN modulates NF-kB and AP-1, as well as MAPKs such as ERK1, ERK2, p38α, JNK2, and JNK3.

Our results show that at the colon level of weaned piglets ZEN, at a higher concentration (290 ppb) than the EC recommendation, leads to an increase in the level of NF-kB,

AP-1, and TAK 1, but also at the level of MAPKs, suggesting that ZEN toxicity could be mediated by these molecules, while the exposure to lower concentrations did not induce significant changes.

Another potential mechanism for inducing intestinal inflammation is the activation of the NLRP3 inflammasome. Exposure of intestinal porcine enterocyte cell line (IPEC-J2) cells and mouse peritoneal macrophages from Balb/c mice to ZEN 8 μg/mL led to an increase in IL-1β synthesis, a mechanism presumably mediated by the activation of the NLRP3 inflammasome [54]. However, in the present study in the colon of piglets, exposure to 75 ppb and 290 ppb ZEN did not induce changes in the gene expression of the *NLRP3* inflammasome but led to a significant decrease in the *NLRP6* inflammasome in 290 ppb ZEN. Studies show that the NLRP6 inflammasome plays an important role in the Intestinal microbiome, but also in infectious and inflammatory gastrointestinal diseases. NLRP6 attenuates NF-κB activity, but a decrease in NLRP6 can have a negative effect, stimulating the activation of both the signaling pathway modulated by NF-κB and MAPKs [55]. The results obtained in this study suggest the same effect; simultaneously with the significant decrease in the gene expression of *NLRP6*, an increase in the level of NF-kB, and of MAPKs *ERK1*, *ERK2*, *p38α*, *JNK2*, and *JNK3*, was also observed in the colon of ZEN 290 group. By contrast, ZEN at the concentration below the CE recommendation (75 ppb) has no effect on these signaling molecules.

## 5. Conclusions

The exposure of weaned piglets to a concentration of ZEN below the EU recommendation (ZEN 75) did not induce significant changes in the studied markers at the colon level. Our results show no effect on oxidative stress (protein oxidation, lipid oxidation, antioxidant enzymes activity) and on inflammation markers (IL-1β, TNF-α, IL-8, IFN-γ, IL 4, IL 10), or on the innate immunity markers (TLR2, TLR 4, TLR5, TLR9, MYD88, MD2). Moreover, excepting the increase in gene expression of *TLR4*, *MYD88*, and decrease in antioxidant capacity, the concentration of ZEN above EU recommendation (ZEN 290) did not induce significant changes in innate immunity, oxidative and inflammatory response.

Similarly, the exposure of piglets to ZEN 75 ppb did not induce any changes on the nuclear receptor's genes of *NF-kB*, *AP1*, and *Nrf2*, or MAPKs (*p38α*, *ERK1/2*, *JNK1/2/3*), *NLRP3* and *NLRP6* inflammasome, as well as on other signaling molecules such as *IRAK1*, *TRAF 6*, *TAK 1*, *TGFβ2*, and *AKT*. By contrast, the results obtained when piglets were exposed to ZEN 290 ppb showed that ZEN induced modulation on most molecules involved in signaling pathways involved in inflammation ad oxidative stress. However, more studies are needed at the level of all the organs implicated in immune defense that can be affected by ZEN in order to obtain a complete image necessary to establish a clear guideline regarding the allowed concentration of ZEN in the feed for piglets and to elucidate the underlying mechanisms produced by zearalenone.

**Author Contributions:** Conceptualization, V.C.B., D.E.M. and I.T.; methodology, D.E.M., G.C.P. and I.T.; formal analysis, V.C.B., A.M.P., A.C.A. and I.A.G.; software V.C.B.; investigation, V.C.B., A.M.P., A.C.A. and I.A.G.; resources, D.E.M. and I.T.; writing—original draft preparation, V.C.B.; writing—review and editing, D.E.M. and I.T.; supervision, A.D. and I.T.; project administration, D.E.M. and I.T.; funding acquisition, D.E.M. and I.T. All authors have read and agreed to the published version of the manuscript.

**Funding:** This research was supported by funds from the National Research Projects PCE 42/2022 and 8 PFE/2021 granted by the Romanian Ministry of Research Innovation and Digitalization.

**Institutional Review Board Statement:** The study was conducted in accordance with Romanian law no 206/2004 and approved by the Ethics Committee of the National Institute of Research and Development for Biology and Animal Nutrition.

**Data Availability Statement:** The analyzed sets of data used in the present paper can be offered, on reasonable request, by the corresponding author.

**Conflicts of Interest:** The authors declare no conflict of interest.

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
