# Peer review of "Effects and Underlying Mechanisms of Zearalenone Mycotoxin at Concentrations Close to the EC Recommendation on the Colon of Piglets after Weaning"

_agriculture, doi:10.3390/agriculture13071372_

Round 1
Reviewer 1 Report
Dear Authors
The manuscript is good and had a sound idea that discusses an important topic related to the health of mammals and the effect of zearalenone toxins, as well as studying the limits regulating them. But the next note:
- The number of replicates or experimental animals is very small, and it is difficult to be a general or comprehensive idea through this limited number.
- Ignoring other factors such as bacterial contamination or fungi, whether in the environment surrounding the experimental animals, the water or the feed provided to them.
- Also, not studying other toxins that may be associated with animal feed, such as mycotoxins or bacterial toxins. In fact, it is difficult for zearalenone to be alone toxic on its own without interaction with other secondary metabolites such as trichothecenes, nivalenol, aflatoxin, ochratoxin, DON, or others.
- Although the writing and preparation were done well, the MATERIALS AND METHODS part need more clarification.
- The results need more clear statistical interpretations while, the conclusion needs another wording that matches the objective and/or main basic idea of ​​the manuscript
Author Response
Author response: The Authors want to thank to the Reviewer for the appreciations of the present manuscript and for their comment and observations that will contribute overall to the increase of the manuscript quality.
- Reviewer comment: The manuscript is good and had a sound idea that discusses an important topic related to the health of mammals and the effect of zearalenone toxins, as well as studying the limits regulating them. The number of replicates or experimental animals is very small, and it is difficult to be a general or comprehensive idea through this limited number.
Author response: Thank you for this comment. The present experiment has used a relatively small number of animals (six animals/group, three experimental groups). We agree with the reviewer that the number of individuals in each experimental group could have been bigger, but according to principle of 3Rs concerning the in vivo animal tests that are expensive, time consuming and rise ethical considerations, we have tried to reduce as much as possible the number of animals used in these in vivo toxicological experiments. In the same time, the in vivo experiment was designed to assure a sufficient number of animals/group, in order to allow a fair statistical analyze, each pig being considered as an independent experimental unit.
- Reviewer comment: Ignoring other factors such as bacterial contamination or fungi, whether in the environment surrounding the experimental animals, the water or the feed provided to them. Also, not studying other toxins that may be associated with animal feed, such as mycotoxins or bacterial toxins. In fact, it is difficult for zearalenone to be alone toxic on its own without interaction with other secondary metabolites such as trichothecenes, nivalenol, aflatoxin, ochratoxin, DON, or others.
Author response: Thank you for this comment. Indeed, the co-contamination with more than one toxin or bacteria in food and feed is more frequent than contamination with a single contaminant and this combined effect cannot be neglected. However, in order to assess the effects and mechanisms of action of zearalenone mycotoxin in piglets at concentrations close to the EU recommendation we have to consider only the exposure of piglets to one mycotoxin (in this case zearalenone) as the recommendation refers to individual mycotoxins and not to combinations of mycotoxins or with other toxins (as bacterial toxins). Using ELISA Veratox kits we have analyzed the concentration of other mycotoxins, frequent contaminants of feed (AFB, FB, DON, T2, HT2). Also, as the assurance of food and feed safety is compulsory in human and animal nutrition, we have assessed the microbiological (total number of germs, Escherichia coli, Salmonella) and fungal contamination of experimental feed. All feed formulations felt within the limits allowed by the European legislation in force. This information was now included in the M&M section in the new version of the manuscript:
“Considering the occurrence of mycotoxins naturally in feed, the presence of aflatoxin B1 (AFB1), deoxynivalenol (DON), ochratoxin A (OTA), total fumonisins (B1, B2, B3), and toxin T2 were analyzed by ELISA, using the VERATOX (Neogene, Lansing, MI, USA) kit according to the manufacturer's instructions, with a detection limit between 0.1 and 100ppb (0.5ppb AFB1, 100ppb DON; 1 ppb OTA, 0.2 ppm FBs, 5 ppb ZEN). The concentration of all investigated mycotoxins was below the detection limit. The diets were also screened for contamination with bacteria (total bacteria count, Escherichia coli, Salmonella) and with fungus (total fungi count) and the contamination levels were found to be under the EU limits accepted for pigs”.
- Reviewer comment: Although the writing and preparation were done well, the MATERIALS AND METHODS part need more clarification.
Author response: Thank you for this comment. As suggested, more details concerning the experimental design and the different techniques used in the present paper were added in the new version of the manuscript.
- Reviewer comment: The results need more clear statistical interpretations while, the conclusion needs another wording that matches the objective and/or main basic idea of ​​the manuscript
Author response: Thank you for this comment. As suggested, in order to improve the statistical interpretation, we have revised the statistical differences of the results and additional details such as the significant trend and P- value were added. Also, as suggested, we have modified the conclusion section in the new version of the manuscript.
Corrections were made to the minor remarks in the entire manuscript. All changes from the revised version of our manuscript appear in red.

Reviewer 2 Report
Some recommendations to improve the manuscript and the research are the following:
Line 73-85. Exist some reference for this method or is it from this research?
Line 99: Change ARN --> RNA
Table 1. Place the primers directionality (5'.....3')
Line 140: Define PBS
Line 155: 10 000 g --> 10 000 x g
ml --> mL
Line 330: Full name of E. coli.
Line 337: Define DON
Line 387: Define PBMC
The research would have more impact if you conducted the same experiments with a group that increased the ZEN concentration to 350 ppb. This would make it more conclusive and support the conclusion about the low toxicity of ZEN within the limits proposed by the EC.
Author Response
Author response: The Authors want to thank to the Reviewer for the appreciations of the present manuscript and for their comment and observations that will contribute overall to the increase of the manuscript quality.
Reviewer #2 Comments:
Some recommendations to improve the manuscript and the research are the following:
- Comment: Line 73-85. Exist some reference for this method or is it from this research?
The experimental design is for this study and has been used in other studies by the team (Marin et al., 2018; Pistol et al., 2020, 2021; Taranu et al., 2020).
Marin, D. E., Pistol, G. C., Gras, M., Palade, M., & Taranu, I. (2018). A comparison between the effects of ochratoxin A and aristolochic acid on the inflammation and oxidative stress in the liver and kidney of weanling piglets. Naunyn-Schmiedeberg’s Archives of Pharmacology, 391(10), 1147–1156. https://doi.org/10.1007/s00210-018-1538-9
Pistol, G. C., Bulgaru, C. V., Marin, D. E., Oancea, A. G., & Taranu, I. (2021). Dietary Grape Seed Meal Bioactive Compounds Alleviate Epithelial Dysfunctions and Attenuates Inflammation in Colon of DSS-Treated Piglets. Foods, 10(3). https://doi.org/10.3390/foods10030530
Pistol, G. C., Marin, D. E., Rotar, M. C., Ropota, M., & Taranu, I. (2020). Bioactive compounds from dietary whole grape seed meal improved colonic inflammation via inhibition of MAPKs and NF-kB signaling in pigs with DSS induced colitis. Journal of Functional Foods, 66, 103708. https://doi.org/https://doi.org/10.1016/j.jff.2019.103708
Taranu, I., Hermenean, A., Bulgaru, C., Pistol, G. C., Ciceu, A., Grosu, I. A., & Marin, D. E. (2020). Diet containing grape seed meal by-product counteracts AFB1 toxicity in liver of pig after weaning. Ecotoxicology and Environmental Safety, 203, 110899. https://doi.org/https://doi.org/10.1016/j.ecoenv.2020.110899
- Comment: Line 99: Change ARN --> RNA
As recommended, the correction was done.
- Comment:Table 1. Place the primers directionality (5'.....3')
As recommended, the primer’s directionality from Table 1 was changed from fw to 5’-3’ and from rv to 3’- 5’.
- Comment: Line 140: Define PBS
As recommended, PBS was defined as phosphate-buffered saline.
- Comment: Line 155: 10 000 g --> 10 000 x g
As recommended, the correction was done.
- Comment: ml --> mL
As recommended, the correction was done, and all the measurement units were verified and corrected I the new version of the manuscript.
- Comment: Line 330: Full name of E. coli.
As recommended, the full name of E.coli was added in the text as Escherichia coli.
- Comment: Line 337: Define DON
As recommended, DON was defined as deoxynivalenol.
- Comment: Line 387: Define PBMC
As recommended, PBMC was defined in the text as peripheral blood mononuclear cell.
- Comment: The research would have more impact if you conducted the same experiments with a group that increased the ZEN concentration to 350 ppb. This would make it more conclusive and support the conclusion about the low toxicity of ZEN within the limits proposed by the EC.
Indeed, an increased concentration of ZEA would have had significant results. In another study, our team investigated the effect of a higher concentration of ZEN (316 ppb) in pigs after weaning and found a significant dramatically decrease in cytokines and other inflammatory markers (Pistol et al., 2014). Compared to control, ZEN contaminated diet induced significant changes on global transcriptome in spleen Pistol et al., 2015)
In the present study we were more interested in the effects of the concentration lower than the EU recommendation (100 ppb). The dose of 290 ppb served for comparison. That was the reason we chose it.
Corrections were made to the minor remarks in the entire manuscript. All changes from the revised version of our manuscript appear in red.

Reviewer 3 Report
The manuscript described the effects of ZEN in concentrations below (75ppb) and
above (290ppb) EU recommendation on the level of some key markers involved in the oxidative and inflammatory response, as well as the mechanisms and signaling pathways, through which ZEN could produce its toxicity in the colon of weaned piglets.
The results showed that the EC recommendation can be applied as a norm, the exposure of the piglets’ colon to the lower concentration of ZEN (75ppb) did not lead to changes in stress and inflammation markers, nor in the signaling pathways associated with these processes. The data were a useful complement to the EC recommendation standard, which was a meaningful work.
However, it has some questions to be answered before it is accepted.
1. Figure 4 contained a and b, which should be described in the text, respectively.
2. There were obvious differences among the Nrf2 of the control sample and ZEN75 and ZEN 90. Why did authors described there were not a significant effect caused by any of the analyzed concentrations in line 221-222.
The language should be improved.
Author Response
Author response: The Authors want to thank to the Reviewer for the appreciations of the present manuscript and for their comment and observations that will contribute overall to the increase of the manuscript quality.
Reviewer #3 Comments:
The manuscript described the effects of ZEN in concentrations below (75ppb) and above (290ppb) EU recommendation on the level of some key markers involved in the oxidative and inflammatory response, as well as the mechanisms and signaling pathways, through which ZEN could produce its toxicity in the colon of weaned piglets.
The results showed that the EC recommendation can be applied as a norm, the exposure of the piglets’ colon to the lower concentration of ZEN (75ppb) did not lead to changes in stress and inflammation markers, nor in the signaling pathways associated with these processes. The data were a useful complement to the EC recommendation standard, which was meaningful work.
However, it has some questions to be answered before it is accepted.
- Comment: Figure 4 contained a and b, which should be described in the text, respectively.
As recommended, Figure 4 was described in the text as follows: “Analyzing the gene (Figure 4a) and protein expression (Figure 4b) of the Nrf2 signaling molecule…”
- Comment: There were obvious differences between the Nrf2 of the control sample and ZEN75 and ZEN 290. Why did authors describe there were not a significant effect caused by any of the analyzed concentrations in line 22
Indeed, in Figure 4 b) there is a difference between the three experimental groups, but it was not statistically significant. The very high biological variability and the small number of animals may be the reason for which the difference is not significant. However, after a statistical reanalysis an increasing trend was observed in Nrf2 protein expression in the case of ZEN 75: Control vs ZEN 75 (p= 0.0868), but not for Control vs ZEN 290 (p=0.8986). These details were added in the new version of the manuscript, lines 228-229.
Corrections were made to the minor remarks in the entire manuscript. All changes from the revised version of our manuscript appear in red.

Reviewer 4 Report
Effects and mechanisms of action of zearalenone mycotoxin in piglet’s colon, at concentrations close to the EC recommendation (Awkward title as it implies the colon is at concentrations close to EC recommendation vs the zearalenone mycotoxin)
The authors need to explain/ justify why the colon was chosen to evaluate intestinal health for this paper. On lines 57 and 58 the authors refer to the.. Gut, and especially the colon represents the most vulnerable tissue to inflammation and oxidative stress induced by weaning (10)… But I (and much literature) challenge that statement – the small intestine is generally regarded as the most sensitive to diet changes and microbiota challenges, while the colon is regarded as an organ that absorbs water and some nutrients. Also- Reference #10 - Cheng W, Jiang S, Huang L et al. Effects of zearalenone-induced oxidative stress and Keap1-Nrf2 signaling pathway-related gene expression in the ileum and mesenteric lymph nodes of post-weaning gilts. Toxicology 2020.;429:152337 uses the ileum or small intestine and I did not find one mention of the colon – erroneous citation.
[Additional information see also: Tang X, Kangning X, Fang R, Li M. Weaning stress and intestinal health of piglets: A review. Doi 10.3389/fimmu.2022.1042778]
Line 32 …, T-2 and HT-2 toxins, and … - if you are going to use the word ‘toxins’ you need to add ‘and’ and omit the comma.
Line 32 spelling it is ‘fumonisins’ and not with a z; I would recommend just using fumonisins or use fumonisins B1 and B2 (B3 is not used in the EC guidelines but is used in the US FDA guidelines).
Line 33.. wording…Farm animals are often affected by mycotoxins, particularly the swine species due to the high cereal content in the diet and their native sensitivity (3)….
Line 39 … digestive, immune, and nervous system (5,6).
Lines 67-68 consider .. until final concentrations of 75 ug/kg feed (ppb or part per billion) and 290 ug/kg feed or ppb were obtained.
Line 70 clarify if that was testing for fumonisins B1 and B2 (?B3)
Line 71 and 72 English… The concentrations of all investigated mycotoxins were below the detection limits. As an interested reader – what were the detection limits of the test for the various mycotoxins since I don’t use Veratox?
Section 2.2
Please provide the age or weight of the weaned pigs. Also, the ingredients in the basal diet should be included for the reader’s knowledge.
Line 100 …. Protocol described by Marin et al. [ 5]… Ref #5 is Jia et al. Ref #12 is Marin et al. (shows up on line 142) – but you will need to change reference numbers in the paper and references because Ref #11 Pistol et al. is noted on line 122.
Line 165 Define OD(S) and OD(P) – please define for the reader.
Figure 3 and lines 209 and 210 for TAC in the Figure 3 a significant effect was indicated with *** but no p-value was given in the text.
Lines 235 to 242stated … not affect significantly the inflammatory response … IL-1βin Figure 6 BUT IN Figure 6 IL-1β there is a * for significant effect for ZEN290?? That was described as a higher level of expression but insignificant.
Lines 250 to 252 – results were described and presented in a figure on the effect but identification of ‘Figure 7’ was not acknowledged in the text.
Line 301 – referred to zearalenone with … toxic effects have been observed in the nervous, hepatic, and digestive systems (19). WRONG – ref 19 is Stoev et al paper on ochratoxin A and fumonisin B1.
Line 307 …. The health problems that ZEN-contaminated feed consumption can have on pigs, supplementary studies are necessary … The authors have not demonstrated or supported this broad statement for a regulation vs recommendation.
Lines 311 to 318 – Why did the authors study the colon or large intestine? What led to this decision?
Line 330 … IPEC 1 shown that combined exposure to ZEN …
Line 340 … very few are carried out on pigs or pig cells.
Lines 351 to 353 Sentence structure odd. Try: In vitro tests were performed on porcine Granulosa cells with induction of oxidative stress by low doses of ZEN (15uM, 30 uM) while the mRNA level of the antioxidant enzymes SOD, CAT, and GPx decreased (29).
Line 357 … ZEN at 316 ppb, …
Line 360 .. colon, a very important organ during the weaning period, … Authors have a great deal of study data but as a reader – I only have their statement that the colon is ‘very important during weaning’. This study did not prove the importance of the colon – only of limited effects of low zearalenone concentrations in feed on the colon. Can a weaned pig survive without a colon? It is important in water and electrolyte balance, so it makes sense it is important with a weanling pig undergoing a changing diet and microbial challenges especially with diarrhea. Consider that many humans survive without a colon from cancer treatment.
Line 387 define PBMCs for the reader.
Line 414 What does .. ZEN (6/8ug/ml) fed to … what does 6/8ug/ml mean??
Line 438 to 442 is a very long sentence that can be broken into two sentences.
Lines 446 to 448 Awkward wording. Try: Our study aimed to evaluate the effects at the intestinal level in weaned piglets exposed to ZEN concentrations in feed near the zearalenone limit recommended by the European Commission. For this purpose, weaned piglets have received feed contaminated ….
Lines 452 to 453 – the authors have not proved that 100 ppb ZEN in feed is ‘safe’ for piglets – no clinical pathology, gross or histopathology, or performance data was included in this study so all that the authors can conclude is that ..Based on the results of this study, feeding weaned pigs a concentration of ZEN below the EU recommendation limit of 100 ppb could (or can) be considered safe (or innocuous) for the pigs.
Line 458 … oxidative and inflammatory responses.
Line 462 … proved to be safe as it … Again, I would be cautious using the word ‘safe’ due to the limited nature of this study and say that Zen 75 ppb did not induce any changes at the level of nuclear receptors … (and mention) no significant effects on the oxidative and inflammatory responses.
Line 466 … ZEN 290ppb (similar to the ZEN 75ppb mentioned in line 462) showed that ZEN induced modifications of most of the markers evaluated in this study.
Line 466-467 (.. most of the markers). Additional studies are necessary to elucidate the mechanisms of the action of ZEN. (Add another sentence)
Need to check grammar, punctuation, and English comments. Some of the sentences were long and could be broken into 2 sentences. See attached comments
Author Response
Author response: The Authors want to thank to the Reviewer for the appreciations of the present manuscript and for their comment and observations that will contribute overall to the increase of the manuscript quality.
Reviewer #4 Comments:
- Comment: Effects and mechanisms of action of zearalenone mycotoxin in piglet’s colon, at concentrations close to the EC recommendation (Awkward title as it implies the colon is at concentrations close to EC recommendation vs the zearalenone mycotoxin)
As suggested the title was changed to “Effects and underlying mechanisms of Zearalenone Mycotoxin at concentrations close to the EC Recommendation, on the colon of piglets after weaning”.
- Comment: The authors need to explain/ justify why the colon was chosen to evaluate intestinal health for this paper. On lines 57 and 58 the authors refer to the..
Gut, and especially the colon represents the most vulnerable tissue to inflammation and oxidative stress induced by weaning (10)… But I (and much literature) challenge that statement – the small intestine is generally regarded as the most sensitive to diet changes and microbiota challenges, while the colon is regarded as an organ that absorbs water and some nutrients. Also- Reference #10 - Cheng W, Jiang S, Huang L et al. Effects of zearalenone-induced oxidative stress and Keap1-Nrf2 signaling pathway-related gene expression in the ileum and mesenteric lymph nodes of post-weaning gilts. Toxicology 2020.;429:152337 uses the ileum or small intestine and I did not find one mention of the colon – erroneous citation. [Additional information see also: Tang X, Kangning X, Fang R, Li M. Weaning stress and intestinal health of piglets: A review. Doi 10.3389/fimmu.2022.1042778]
In the introduction, an additional paragraph was added that explains the importance of the colon after weaning:
“During the post-weaning period (1-2 weeks) the gut, including the colon are the most vulnerable tissue (10). Currently, many studies on the exposure of weaned piglets to ZEN are performed at the small intestine level. However, the large intestine takes up undigested feed, therefore, at the level of the colon, the absorption of several substances can take place, apart from water and electrolytes (11). Also, the colon is a gut segment with high microbial activity (active substrate hydrolysis and fermentation). As shown by Richards et al., (2005) colonic microbiota produced fermentation products with protective function stimulating the immune response. But, at weaning several defaults in the intestinal barrier function was observed which could be a starting point for inflammation as well as water and electrolyte imbalance (Boudry et al., 2004). During this period, an increase in secretory activity and permeability was observed in the colon (12). Moreover, diarrhea frequently occurred during the weaning period and is often associated either with the shifts in microbiota (suppression of several beneficial lactic bacteria) after the transition to solid feed or with colonic inflammation caused by enterotoxigenic Escherichia coli or other pathogens (13). These pathogens increased the inflammatory cytokines in the colon contributing to colonic inflammation. Zearalenone can be an additional pro-inflammatory factor that increases colonic inflammation if the feed that the piglets eat during this period is contaminated with ZEN. Bauer et al., (2006) reported that feed type and quality had an important influence on gut microbiota. Reddy et al. (2018) analyzing the colon content of three groups of pigs fed diet contaminated with zearalenone (800ppb) and deoxynivalenol (8000 ppb) reported that the toxins contaminated diet significantly affected the colon microbiota especially Lactobacillus. Similarly, previous study of our team found also that ZEN (290ppb in the diet) decreased the populations of Lactobacillus and Bifidobacterium (Grosu et al., 2023). However, the effects of ZEN on the pig colon microbiota and immune and stress response have not been well-defined.”
- Comment: Line 32 …, T-2 and HT-2 toxins, and … - if you are going to use the word ‘toxins’ you need to add ‘and’ and omit the comma. As recommended, the correction was done.
- Comment: Line 32 spelling it is ‘fumonisins’ and not with a z; I would recommend just using fumonisins or use fumonisins B1 and B2 (B3 is not used in the EC guidelines but is used in the US FDA guidelines). As suggested, the correction was done, and the recommended word fumonisins was used in text.
- Comment: Line 33.. wording…Farm animals are often affected by mycotoxins, particularly the swine species due to the high cereal content in the diet and their native sensitivity (3)…. As suggested, the wording was changed.
- Comment Line 39 … digestive, immune, and nervous system (5,6). As suggested, the wording was changed.
- Comment: Lines 67-68 consider .. until final concentrations of 75 ug/kg feed (ppb or part per billion) and 290 ug/kg feed or ppb were obtained. As recommended the lines were changed to “until final concentrations of 75 µg ZEN/kg feed and 290 µg ZEN /kg feed were obtained”.
- Comment: Line 70 clarify if that was testing for fumonisins B1 and B2 (?B3). The Veratox kit was used to determine the total fumonisins (B1, B2, B3), as suggested this detail was added in the text.
- Comment: Line 71 and 72 English… The concentrations of all investigated mycotoxins were below the detection limits. As an interested reader – what were the detection limits of the test for the various mycotoxins since I don’t use Veratox?
As recommended, the detection limits were added in the text “using the VERATOX (Neogene, Lansing, MI, USA) kit according to the manufacturer's instructions, with a detection limit between 0.1 and 200ppb (0.5ppb AFB1, 100ppb DON; 1 ppb OTA, 0.2ppm FBs, T2/HT-2 25ppb, 5ppb ZEN).”
Section 2.2
- Comment: Please provide the age or weight of the weaned pigs. Also, the ingredients in the basal diet should be included for the reader’s knowledge.
As requested, the age weight of piglets and the basal diet ingredients were added in the Experimental Design section: “The hybrid piglets were weaned at 27 days of age with a body weight of 11.25 ± 1.14 kg. The ingredients of the basal diet were described by Grosu et al., 2023 (11). Briefly the diet contained corn (68.46%), soybean meal (19%), vegetal milk (5%), corn gluten (4 %), l-Lysine (0.31%), Methionine (0.10%), CaCO3, (1.57), Ca (H2PO4)2, mineral-vitamin premix (1%), choline premix (0.10%), phytase (0.01%).”
- Comment: Line 100 …. Protocol described by Marin et al. [ 5] Ref #5 is Jia et al. Ref #12 is Marin et al. (shows up on line 142) – but you will need to change reference numbers in the paper and references because Ref #11 Pistol et al. is noted on line 122. As recommended the bibliography was revised.
- Comment: Line 165 Define OD(S) and OD(P) – please define for the reader.
As recommended, the terms were defined in the Assessment of oxidative stress section as OD- optical density, (P) pellet and (S) supernatant.
- Comment: Figure 3 and lines 209 and 210 for TAC in the Figure 3 a significant effect was indicated with *** but no p-value was given in the text.
As suggested, the p-value was added for TAC, the line was changed in “the case of ZEN 75 (p=0.025) and ZEN 290 (p=0.006)”.
- Comment: Lines 235 to 242stated … not significantly affect the inflammatory response … IL-1βin Figure 6 BUT IN Figure 6 IL-1β there is a * for significant effect for ZEN290?? That was described as a higher level of expression but insignificant.
Indeed, there is a significant increase between ZEN 75 and ZEN 290 in the case of gene expression of IL-1β, but we took into consideration only the comparison between each experimental group reported to the control group.
- Comment: Lines 250 to 252 – results were described and presented in a figure on the effect but identification of ‘Figure 7’ was not acknowledged in the text.
As recommended, the identification of ‘Figure 7’ was added to the results.
- Comment: Line 301 – referred to zearalenone with … toxic effects have been observed in the nervous, hepatic, and digestive systems (19). WRONG – ref 19 is Stoev et al paper on ochratoxin A and fumonisin B1.
After reviewing the bibliography, the reference was changed with Ben Salah-Abbès, J., Belgacem, H., Ezzdini, K., Abdel-Wahhab, M. A., & Abbès, S. (2020). Zearalenone nephrotoxicity: DNA fragmentation, apoptotic gene expression and oxidative stress protected by Lactobacillus plantarum MON03. Toxicon, 175, 28–35. https://doi.org/https://doi.org/10.1016/j.toxicon.2019.12.004
- Line 307 …. The health problems that ZEN-contaminated feed consumption can have on pigs, supplementary studies are necessary … The authors have not demonstrated or supported this broad statement for a regulation vs recommendation.
There are few studies on the effects of 100ppb in feed. Moreover, there are studies on pigs that reported significant effects at a concentration of 100 ppb ZEN in the feed (Braicu et al., 2016) and others in which this concentration had no statistically notable effects (Taranu et al., 2016)
- Comment: Lines 311 to 318 – Why did the authors study the colon or large intestine? What led to this decision?
Many studies showed the importance of the colon for the body. Mainly, the large intestine takes up undigested feed, therefore, at the level of the colon, the absorption of several substances can take place, apart from water and electrolytes. Also, the colon is a gut segment with high microbial activity (active substrate hydrolysis and fermentation). As shown by Richards et al., (2005) colonic microbiota produced fermentation products with protective function stimulating the immune response. As we said above, studies on pigs, on the effects of ZEN on gut (microbiota, immune response or others) are lacking. However, Reddy et al. (2018) analyzing the colon content of three groups of pigs fed diet contaminated with zearalenone (800ppb) and deoxynivalenol (8000 ppb) reported that the toxins contaminated diet significantly affected the colon microbiota especially Lactobacillus. Similarly, previous study of our team found also that ZEN (290ppb in the diet) decreased the populations of Lactobacillus and Bifidobacterium (Grosu et al., 2023).
In pig at weaning the colon is one of the most sensitive and vulnerable organs. Several defaults in the intestinal barrier function were observed which could lead to inflammation as well as to water and electrolyte imbalance (Boudry et al., 2004). Moreover, diarrhea frequently occurred during the weaning period and is often associated either with the shifts in microbiota (suppression of several beneficial lactic bacteria) after the transition to solid feed or with colonic inflammation caused by enterotoxigenic Escherichia coli or other pathogens . These pathogens increased the inflammatory cytokines production in the colon contributing to colonic inflammation. Zearalenone can be an additional pro-inflammatory factor that increases colonic inflammation if the feed during this period is contaminated with ZEN..”
- Comment: Line 330 … IPEC 1 shown that combined exposure to ZEN …
As suggested, the grammar was corrected.
- Comment: Line 340 … very few are carried out on pigs or pig cells.
As suggested, the grammar was corrected.
- Comment: Lines 351 to 353 Sentence structure odd. Try: In vitro tests were performed on porcine Granulosa cells with induction of oxidative stress by low doses of ZEN (15uM, 30 uM) while the mRNA level of the antioxidant enzymes SOD, CAT, and GPx decreased (29).
As recommended, the sentence was replaced.
- Comment: Line 357 … ZEN at 316 ppb, …. As recommended, the grammar was corrected.
- Comment: Line 360 . colon, a very important organ during the weaning period, … Authors have a great deal of study data but as a reader – I only have their statement that the colon is ‘very important during weaning’. This study did not prove the importance of the colon – only of limited effects of low zearalenone concentrations in feed on the colon. Can a weaned pig survive without a colon? It is important in water and electrolyte balance, so it makes sense it is important with a weanling pig undergoing a changing diet and microbial challenges especially with diarrhea. Consider that many humans survive without a colon from cancer treatment.
As we explained above a additional paragraph was added in the introduction in which we showed the importance of the colon during the weaning period. As we said diarrhea frequently occurred during the weaning period and is often associated either with the shifts in microbiota (suppression of several beneficial lactic bacteria) after the transition to solid feed or with colonic inflammation caused by enterotoxigenic Escherichia coli or other pathogens. These pathogens increased the inflammatory cytokines in the colon contributing to colonic inflammation. Furthermore, studies indicate that the inflammation of the intestinal barrier mucosa (colon included) and the increase in its permeability during the piglet`s weaning period can have long-term effects.
Honestly, we don't know if piglets can survive without a colon, because it was not studied, but the problem of weaning, a complex multifactorial problem that is complicated by the interplay of multiple causative agents, host immunity, environmental factors, nutritional factors, and management conditions in young animals cannot be extrapolated / compared with colon cancer in human adults. The problem regards the livestock industry, where the problem is not only animal survival, but obtaining performance from the healthiest animals.
- Comment: Line 387 define PBMCs for the reader. As recommended, PBMC was defined in the text as peripheral blood mononuclear cell.
- Comment: Line 414 What does .. ZEN (6/8ug/ml) fed to … what does 6/8ug/ml mean?? As requested, the clarifications were added “ZEN (6µg/mL or 8µg/mL)”
- Comment: Line 438 to 442 is a very long sentence that can be broken into two sentences. As suggested, the sentence was broken into two smaller sentences.
- Comment: Lines 446 to 448 Awkward wording. Try: Our study aimed to evaluate the effects at the intestinal level in weaned piglets exposed to ZEN concentrations in feed near the zearalenone limit recommended by the European Commission. For this purpose, weaned piglets have received feed contaminated …. As recommended, the wording was replaced with the suggested sentence.
- Comment: Lines 452 to 453 – the authors have not proved that 100 ppb ZEN in feed is ‘safe’ for piglets – no clinical pathology, gross or histopathology, or performance data was included in this study so all that the authors can conclude is that ..Based on the results of this study, feeding weaned pigs a concentration of ZEN below the EU recommendation limit of 100 ppb could (or can) be considered safe (or innocuous) for the pigs.
In the introduction, I added a paragraph in which we explain that the concentration of 100 ppb as the tolerance limit for young piglets is only a recommendation, not a norm, in any case, it is not certain because there are studies that show significant effects at 100 ppb and others that do not reported notable effects at the same concentration.
- Comment: Line 458 … oxidative and inflammatory responses. As recommended, the text was changed.
- Comment: Line 462 … proved to be safe as it … Again, I would be cautious using the word ‘safe’ due to the limited nature of this study and say that Zen 75 ppb did not induce any changes at the level of nuclear receptors … (and mention) no significant effects on the oxidative and inflammatory responses. As recommended, the conclusion section was modified, and the results were summarised at the colon level, the limited nature of the study was taken into consideration.
- Comment: Line 466 … ZEN 290ppb (similar to the ZEN 75ppb mentioned in line 462) showed that ZEN induced modifications of most of the markers evaluated in this study. As recommended, the text was changed.
- Comment: Line 466-467 (most of the markers). Additional studies are necessary to elucidate the mechanisms of the action of ZEN. (Add another sentence).
“Moreover, more studies are needed at the level of all the organs that can be affected by ZEN, in order to obtain a complete picture that could lead to the establishment of clear guidelines regarding the presence of ZEN in the feed intended for piglets.”
Corrections were made to the minor remarks in the entire manuscript. All changes from the revised version of our manuscript appear in red.

Round 2
Reviewer 3 Report
The authors have revised the corresponding sections and it can be accepted.